# A NEW POINTWISE CONVOLUTION IN DEEP NEURAL NETWORKS THROUGH EXTREMELY FAST AND NON PARAMETRIC TRANSFORMS

## ABSTRACT

Some conventional transforms such as Discrete Walsh-Hadamard Transform (DWHT) and Discrete Cosine Transform (DCT) have been widely used as feature extractors in image processing but rarely applied in neural networks. However, we found that these conventional transforms have the ability to capture the cross-channel correlations without any learnable parameters in DNNs. This paper firstly proposes to apply conventional transforms on pointwise convolution, showing that such transforms significantly reduce the computational complexity of neural networks without accuracy degradation. Especially for DWHT, it requires no floating point multiplications but only additions and subtractions, which can considerably reduce computation overheads. In addition, its fast algorithm further reduces complexity of floating point addition from $\mathcal{O}(n^2)$ to $\mathcal{O}(n \log n)$. These non-parametric and low computational properties construct extremely efficient networks in terms of the number of parameters and operations, enjoying accuracy gain. Our proposed DWHT-based model gained 1.49% accuracy increase with 79.4% reduced parameters and 49.4% reduced FLOPs compared with its baseline model (MoblieNet-V1) on the CIFAR 100 dataset.

## 1 INTRODUCTION

Large Convolutional Neural Networks (CNNs) (Krizhevsky et al., 2012; Simonyan & Zisserman, 2014; He et al., 2016; Szegedy et al., 2016b;a) and automatic Neural Architecture Search (NAS) based networks (Zoph et al., 2018; Liu et al., 2018; Real et al., 2018) have evolved to show remarkable accuracy on various tasks such as image classification (Deng et al., 2009; Krizhevsky & Hinton, 2009), object detection (Lin et al., 2014), benefited from huge amount of learnable parameters and computations. However, these large number of weights and high computational cost enabled only limited applications for mobile devices that require the constraint on memory space being low as well as for devices that require real-time computations (Canziani et al., 2016).

With regard to solving these problems, Howard et al. (2017); Sandler et al. (2018); Zhang et al. (2017b); Ma et al. (2018) proposed parameter and computation efficient blocks while maintaining almost same accuracy compared to other heavy CNN models. All of these blocks utilized depthwise separable convolution, which deconstructed the standard convolution with the $(3 \times 3 \times C)$ size for each kernel into spatial information specific depthwise convolution $(3 \times 3 \times 1)$ and channel information specific pointwise $(1 \times 1 \times C)$ convolution. The depthwise separable convolution achieved comparable accuracy compared to standard spatial convolution with hugely reduced parameters and FLOPs. These reduced resource requirements made the depthwise separable convolution as well as pointwise convolution (PC) more widely used in modern CNN architectures.

Nevertheless, we point out that the existing PC layer is still computationally expensive and occupies a lot of proportion in the number of weight parameters (Howard et al., 2017). Although the demand toward PC layer has been and will be growing exponentially in modern neural network architectures, there has been a little research on improving the naive structure of itself.

Therefore, this paper proposes a new PC layer formulated by non-parametric and extremely fast conventional transforms. Conventional transforms that we applied on CNN models are Discrete

Walsh-Hadamard Transform (DWHT) and Discrete Cosine Transform (DCT), which have widely been used in image processing but rarely been applied in CNNs (Ghosh & Chellappa, 2016).

We empirically found that although both of these transforms do not require any learnable parameters at all, they show the sufficient ability to capture the cross-channel correlations. This non-parametric property enables our proposed CNN models to be significantly compressed in terms of the number of parameters, leading to get the advantages (i.e. efficient distributed training, less communication between server and clients) referred by Iandola et al. (2016). We note that especially DWHT is considered to be a good replacement of the conventional PC layer, as it requires no floating point multiplications but only additions and subtractions by which the computation overheads of PC layers can significantly be reduced. Furthermore, DWHT can take a strong advantage of its fast version where the computation complexity of the floating point operations is reduced from $\mathcal{O}(n^2)$ to $\mathcal{O}(n \log n)$. These non-parametric and low computational properties construct extremely efficient neural network from the perspective of parameter and computation as well as enjoying accuracy gain.

Our contributions are summarized as follows:

- We propose a new PC layer formulated with conventional transforms which do not require any learnable parameters as well as significantly reducing the number of floating point operations compared to the existing PC layer.

- The great benefits of using the bases of existing transforms come from their fast versions, which drastically decrease computation complexity in neural networks without degrading accuracy.

- We found that applying ReLU after conventional transforms discards important information extracted, leading to significant drop in accuracy. Based on this finding, we propose the optimal computation block for conventional transforms.

- We also found that the conventional transforms can effectively be used especially for extracting high-level features in neural networks. Based on this, we propose a new transform-based neural network architecture. Specifically, using DWHT, our proposed method yields 1.49% accuracy gain as well as 79.4% and 49.4% reduced parameters and FLOPs, respectively, compared with its baseline model (MobileNet-V1) on CIFAR 100 dataset.

## 2 RELATED WORK

### 2.1 DECONSTRUCTION AND DECOMPOSITION OF CONVOLUTIONS

For reducing computation complexity of the existing convolution methods, several approaches of rethinking and deconstructing the naive convolution structures have been proposed. Simonyan & Zisserman (2014) factorized a large sized kernel (e.g. $5 \times 5$) in a convolution layer into several small size kernels (e.g. $3 \times 3$) with several convolution layers. Jeon & Kim (2017) pointed out the limitation of existing convolution that it has the fixed receptive field. Consequently, they introduced learnable spatial displacement parameters, showing flexibility of convolution. Based on Jeon & Kim (2017), Jeon & Kim (2018) proved that the standard convolution can be deconstructed as a single PC layer with the spatially shifted channels. Based on that, they proposed a very efficient convolution layer, namely active shift layer, by replacing spatial convolutions with shift operations.

It is worth noting that the existing PC layer takes the huge proportion of computation and the number of weight parameters in modern lightweight CNN models (Howard et al., 2017; Sandler et al., 2018; Ma et al., 2018). Specifically, MobileNet-V1 (Howard et al., 2017) requires 94%, 74% of the overall computational cost and the overall number of weight parameters for the existing PC layer, respectively. Therefore, there were attempts to reduce computation complexity of PC layer. Zhang et al. (2017b) proposed ShuffleNet-V1 where the features are decomposed into several groups over channels and PC operation was conducted for each group, thus reducing the number of weight parameters and FLOPs by the number of groups $G$. However, it was proved in Ma et al. (2018) that the memory access cost increases as $G$ increases, leading to slower inference speed. Similarly to the aforementioned methods, our work is to reduce computation complexity and the number of weight parameters in a convolution layer. However, our objective is more oriented on finding out mathe-

matically efficient algorithms that make the weights in convolution kernels more effective in feature representation as well as efficient in terms of computation.

## 2.2 QUANTIZATION

Quantization in neural networks reduced the number of bits utilized to represent the weights and/or activations. Vanhoucke et al. (2011) applied 8-bit quantization on weight parameters, which enabled considerable speed-up with small drop of accuracy. Gupta et al. (2015) applied 16-bit fixed point representation with stochastic rounding. Based on Han et al. (2015b) which pruned the unimportant weight connections through thresholding the values of weight, Han et al. (2015a) successfully combined the pruning with 8 bits or less quantization and huffman encoding. The extreme case of quantized networks was evolved from Courbariaux et al. (2015), which approximated weights with the binary $(+1, -1)$ values. From the milestone of Courbariaux et al. (2015), Courbariaux & Bengio (2016); Hubara et al. (2016) constructed Binarized Neural Networks which either stochastically or deterministically binarize the real value weights and activations. These binarized weights and activations lead to significantly reduced run-time by replacing floating point multiplications with 1-bit XNOR operations.

Based on Binarized Neural Networks (Courbariaux & Bengio, 2016; Hubara et al., 2016), Local Binary CNN (Juefei-Xu et al., 2016) proposed a convolution module that utilizes binarized non-learnable weights in spatial convolution based on Local Binary Patterns, thus replacing multiplications with addition/subtraction operations in spatial convolution. However, they did not consider reducing computation complexity in PC layer and remained the weights of PC layer learnable floating point variables. Our work shares the similarity to Local Binary CNN (Juefei-Xu et al., 2016) in using binary fixed weight values. However, Local Binary Patterns have some limitations for being applied in CNN since they can only be used in spatial convolution as well as there are no approaches that enable fast computation of them.

## 2.3 CONVENTIONAL TRANSFORMS

In general, several transform techniques have been applied for image processing. Discrete Cosine Transform (DCT) has been used as a powerful feature extractor (Dabbaghchian et al., 2010). For $N$-point input sequence, the basis kernel of DCT is defined as a list of cosine values as below:

$$C_m = [cos(\frac{(2x+1)m\pi}{2N})], \qquad 0 \le x \le N - 1 \tag{1}$$

where $m$ is the index of a basis and captures higher frequency information in the input signal as $m$ increases. This property led DCT to be widely applied in image/video compression techniques that emphasize the powers of image signals in low frequency regions (Rao & Yip, 2014).

Discrete Walsh Hadamard Transform (DWHT) is a very fast and efficient transform by using only $+1$ and $-1$ elements in kernels. These binary elements in kernels allow DWHT to perform without any multiplication operations but addition/subtraction operations. Therefore, DWHT has been widely used for fast feature extraction in many practical applications, such as texture image segmentation (Vard et al., 2011), face recognition (Hassan et al., 2007), and video shot boundary detection (G. & S., 2014).

Further, DWHT can take advantage of a structured-wiring-based fast algorithm (Algorithm 1) as well as allowing very high efficiency in encoding the spatial information (Pratt et al., 1969). The basis kernel matrix of DWHT is defined using the previous kernel matrix as below:

$$H^D = \begin{pmatrix} H^{D-1} & H^{D-1} \\ H^{D-1} & -H^{D-1} \end{pmatrix}, \tag{2}$$

where $H^0 = 1$ and $D \ge 1$. In this paper we denote $H_m^D$ as the $m$-th row vector of $H^D$ in Eq. 2. Additionally, we adopt fast DWHT algorithm to reduce computation complexity of PC layer in neural networks, resulting in an extremely fast and efficient neural network.

## 3 METHOD

We propose a new PC layer which is computed with conventional transforms. The conventional PC layer can be formulated as follows:

$$Z_{ijm} = W_m^\top \cdot X_{ij}, \qquad 1 \leq m \leq M \tag{3}$$

where $(i, j)$ is a spatial index, and $m$ is output channel index. In Eq. 3, $N$ and $M$ are the number of input and output channels, respectively. $X_{ij} \in \mathcal{R}^N$ is a vector of input $X$ at the spatial index $(i, j)$, $W_m \in \mathcal{R}^N$ is a vector of $m$-th weight $W$ in Eq. 3. For simplicity, the stride is set as 1 and the bias is omitted in Eq. 3.

Our proposed method is to replace the learnable parameters $W_m$ with the bases in the conventional transforms. For example, replacing $W_m$ with $H_m^D$ in Eq. 3, we now can formulate the new multiplication-free PC layer using DWHT. Similarly, the DCT basis kernels $C_m$ in Eq. 1 can substitute for $W_m$ in Eq. 3, formulating another new PC layer using DCT. Note that the normalization factors in the conventional transforms are not applied in the proposed PC layer, because Batch Normalization (Ioffe & Szegedy, 2015) performs a normalization and a linear transform which can be viewed as a normalization in the existing transforms.

The most important benefit of the proposed method comes from the fact that the fast algorithms of the existing transforms can be applied in the proposed PC layers for further reduction of computation. Directly applying above new PC layer gives computational complexity of $\mathcal{O}(N^2)$. Adopting the fast algorithms, we can significantly reduce the computation complexity of PC layer from $\mathcal{O}(N^2)$ to $\mathcal{O}(N log N)$ without any change of the computation results.

We demonstrate the pseudo-code of our proposed fast PC layer using DWHT in Algorithm 1 based on the Fast DWHT structure shown in Figure 1a. In Algorithm 1, for $log N$ iterations, the even-indexed channels and odd-indexed channels are added and subtracted in element-wise manner, respectively. The resulting elements which were added and subtracted are placed in the first $N/2$ elements and the last $N/2$ elements of the input of next iteration, respectively. In this computation process, each iteration requires only $N$ operations of additions or subtractions. Consequently, Algorithm 1 gives us complexity of $\mathcal{O}(N log N)$ in addition or subtraction. Compared to the existing PC layer that requires complexity of $\mathcal{O}(N^2)$ in multiplication, our method is extremely cheaper than the conventional PC layer in terms of computation costs as shown in Figure 1b and in power consumption of computing devices (Horowitz, 2014). Note that, similarly to fast DWHT, DCT can also be computed in a fast manner that recursively decomposes the $N$-point input sequence into two subproblems of $N/2$-point DCT (Kok, 1997).

Compared to DWHT, DCT takes advantage of using more natural shapes of cosine basis kernels, which tend to provide better feature extraction performance through capturing the frequency information. However, DCT inevitably needs multiplications for inner product between $C$ and $X$ vectors, and a look up table (LUT) for computing cosine kernel bases which can increase the processing time and memory access. On the other hand, as mentioned, the kernels of DWHT consist only of $+1, -1$ which allows for building a multiplication-free module. Furthermore, any memory access towards kernel bases is not needed if our structured-wiring-based fast DWHT algorithm (Algorithm 1) is applied. Our comprehensive experiments in Section 3.1 and 3.2 show that DWHT is more efficient than DCT in being applied in PC layer in terms of trade-off between the complexity of computation cost and accuracy.

Note that, for securing more general formulation of our newly defined PC layer, we padded zeros along the channel axis if the number of input channels is less than that of output channels while truncating the output channels when the number of output channels shrink compared to that of input channels as shown in Algorithm 1.

Figure 1a shows the architecture of fast DWHT algorithm described in Algorithm 1. This structured-wiring-based architecture ensures that the receptive field of each output channels is $N$, which means each output channel is fully reflected against all input channels through $log_2 N$ iterations. This fully-reflected property helps to capture the input channel correlations in spite of the computation process of what channel elements will be added and subtracted being structured in a deterministic manner.

For successfully fusing our new PC layer into neural networks, we explored two themes: i) an optimal block search for the proposed PC; ii) an optimal insertion strategy of the proposed block found by i), in a hierarchical manner on the blocks of networks. We assumed that there are an optimal block unit structure and an optimal hierarchy level (high-, middle-, low-level) blocks in the neural networks favored by these non-learnable transforms. Therefore, we conducted the experiments for the two aforementioned themes accordingly. We evaluated the effectiveness for each of our networks by accuracy fluctuation as the number of learnable weight parameters or FLOPs changes. For comparison, we counted total FLOPs with summation of the number of multiplications, additions and subtractions performed during the inference. Unless mentioned, we followed the default experimental setting as 128 batch size, 200 training epochs, 0.1 initial learning rate where 0.94 is multiplied per 2 epochs, and 0.9 momentum with 5e-4 weight decay value. In all the experiments, the model accuracy was obtained by taking an average of Top-1 accuracy values from three independent training results.

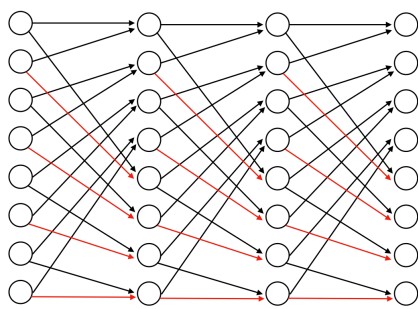

(a) A black circle indicates a channel element, and black and red lines are additions and subtractions, respectively. The number of input channels is set as 8 for simplicity. Best viewed in color.

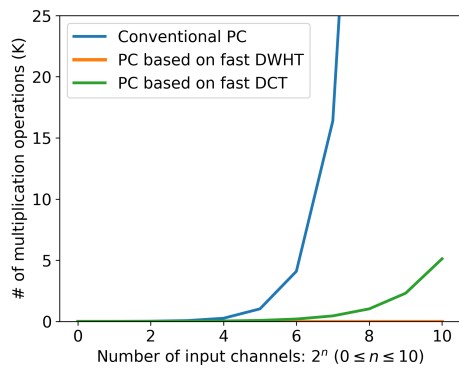

(b) $x$ axis denotes logarithm of the number of input channels which range from $2^0$ to $2^n$. For simplicity, the number of output channels is set to be same as that of the input channel for all PC layers. Best viewed in color.

Figure 1: Left: architecture of our PC layer based on fast DHWT algorithm in Algorithm 1, Right: comparison of the number of multiplications between our new PC layers and the conventional PC layer.

---

**Algorithm 1** A new pointwise convolution using fast DWHT algorithm

**Input:** 4D input features $X(B \times N \times H \times W)$, output channel $M$

1: $n \leftarrow \log_2 N$
2: **if** $N < M$ **then**
3:     ZeroPad1D($X$, axis=1)                    ▷ pad zeros along the channel axis
4: **end if**
5: **for** $i \leftarrow 1$ to $n$ **do**
6:     $e \leftarrow X[:, :: 2, :, :]$
7:     $o \leftarrow X[:, 1 :: 2, :, :]$
8:     $X[:, : N/2, :, :] \leftarrow e + o$
9:     $X[:, N/2 :, :, :] \leftarrow e - o$
10: **end for**
11: **if** $N > M$ **then**
12:     $X \leftarrow X[:, : M, :, :]$
13: **end if**

---

### 3.1 Optimal Block structure for the conventional transforms

From a microscopic perspective, the block unit is the basic foundation of neural networks, and it determines the efficiency of the weight parameter space and computation costs in terms of accuracy. Accordingly, to find the optimal block structure for our proposed PC layer, our experiments are conducted to replace the existing PC layer blocks with our new PC layer blocks in ShuffleNet-V2 (Ma et al., 2018). The proposed block and its variant blocks are listed in Figure 2. Comparing the results of (c) and (d) in Table 1 informs us the important fact that the ReLU (Nair & Hinton, 2010) activation function significantly harms the accuracy of our neural networks equipped with the conventional transforms. We empirically analyzed this phenomenon in Section 4.1. Additionally, comparing the accuracy results of (b) and (d) in Table 1 denotes that the proposed PC layers are superior to the PC layer which randomly initialized and fixed its weights to be non-learnable. These results imply that DWHT and DCT kernels can better extract meaningful information of cross-channel correlations compared to the kernels which are randomly initialized and non-learnable. Compared to the baseline model in Table 1, (d)-DCT w/o ReLU and (d)-DWHT w/o ReLU blocks show accuracy drop by approximately 2.3% under the condition that 42% and 49.5% of learnable weight parameters and FLOPs are reduced, respectively. These results imply that the proposed blocks (c) and (d) are still inefficient in trade-off between accuracy and computation costs of neural networks, leading us to more explore to find out an optimal neural network architecture. In the next subsection, we address this problem through applying conventional transforms on the optimal hierarchy level features (See Section 3.2). Based on our comprehensive experiments, we set the block structure (d) as our default proposed block which will be exploited in all the following experiments.

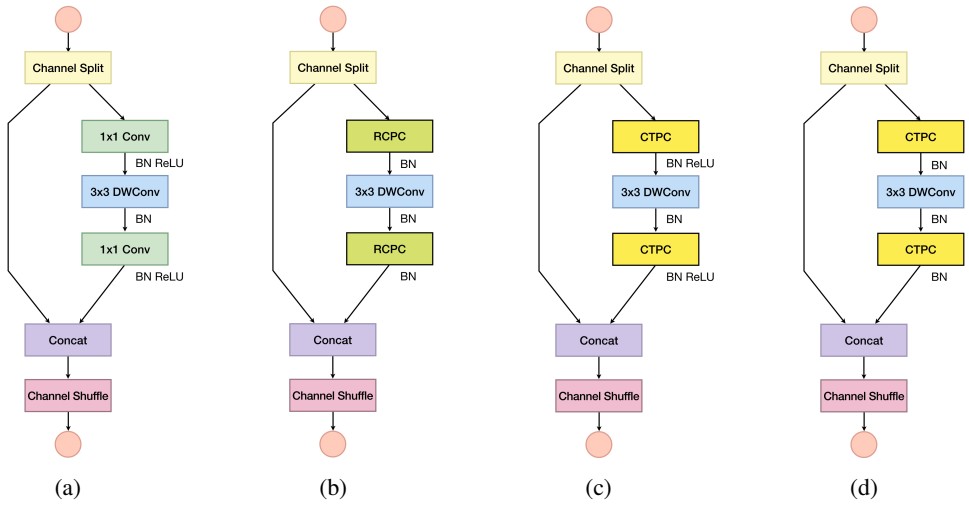

Figure 2: Structure of block units. (a): the basic block of ShuffleNet-V2; (b): block using random constant pointwise convolution (RCPC) layers; (c): block using conventional transform pointwise convolution (CTPC) layers with ReLU applied after each of CTPC layer; (d): our proposed block using CTPC layers without ReLU. Block (b) randomly initialized the weights of PC layer with the distribution of $\mathcal{U}(-1/\sqrt{N/2}, 1/\sqrt{N/2})$, where $N$ is the number of input channel and fixed these weights during training.

### 3.2 Optimal hierarchy level blocks for conventional transforms

In this section, we search on an optimal hierarchy level where our optimal block which is based on the proposed PC layer is effectively applied in a whole network architecture. The optimal hierarchy level will allow the proposed network to have the minimal number of learnable weight parameters and FLOPs without accuracy drop, which is made possible by non-parametric and extremely fast conventional transforms. It is noted that applying our proposed block on the high-level blocks in the network provides much more reduced number of parameters and FLOPs rather than applying on low-level blocks, because channel depth increases exponentially as the layer goes deeper in the network.

| Model | Top-1 Acc (%) | # of Weights (ratio) | # of FLOPs (ratio) |
|---|---|---|---|
| (a)-baseline | $71.68 \pm 0.26$ | 1.57M (1x) | 102.9M (1x) |
| (b)-RCPC w/o ReLU | $68.16 \pm 0.07$ | 0.92M (0.58x) | 102.9M (1x) |
| (c)-DWHT w/ ReLU | $66.2 \pm 0.22$ | 0.92M (0.58x) | 50.2M (0.48x) |
| (c)-DCT w/ ReLU | $66.55 \pm 0.5$ | 0.92M (0.58x) | 54.7M (0.53x) |
| (d)-DWHT w/o ReLU | $69.42 \pm 0.31$ | 0.92M (0.58x) | 50.2M (0.48x) |
| (d)-DCT w/o ReLU | $69.23 \pm 0.14$ | 0.92M (0.58x) | 54.7M (0.53x) |

Table 1: Performance result of block units in Figure 2 on CIFAR100 dataset. All the experimented models are based on ShuffleNet-V2 with width hyper-parameter 1.1x which we customized to make the number of output channels in Stage2, 3, 4 as 128, 256, 512, respectively for fair comparison with DWHT which requires $2^n$ input channels. We replaced all of 13 stride 1 basic blocks (i.e. (a) block) in baseline model with (b), (c), (d) blocks, respectively. (c)-DWHT w/ ReLU denotes CTPC layer in (c) block is based on DWHT, while (d)-DCT w/o ReLU denotes CTPC layer in (d) block is based on DCT.

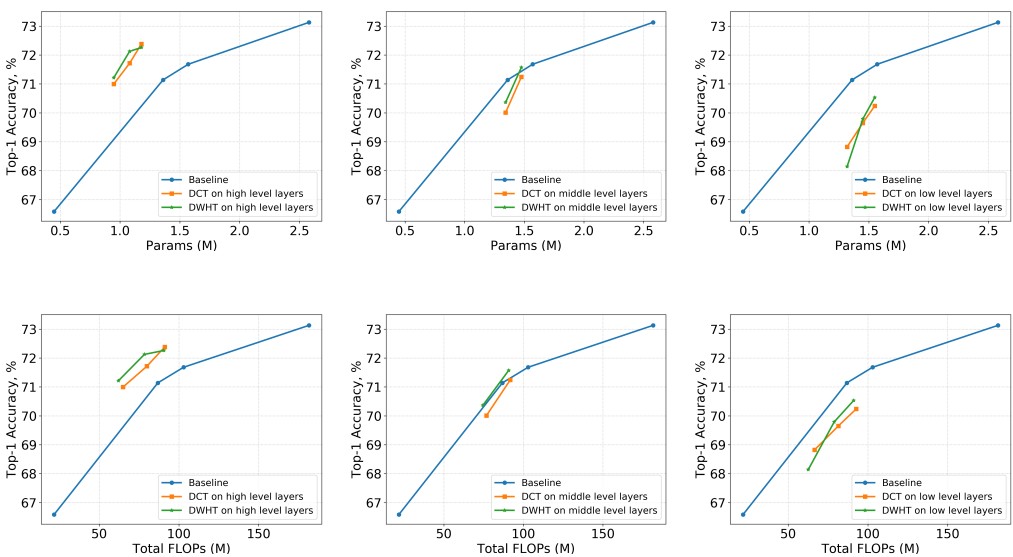

Figure 3: Performance curve of hierarchically applying our optimal block on CIFAR100, Top: in the viewpoint of the number of learnable weight parameters, Bottom: in the viewpoint of the number of FLOPs. The performance of baseline models was evaluated by ShuffleNet-V2 architecture with width hyper-parameter 0.5x, 1x, 1.1x, 1.5x. Our models were all experimented with 1.1x setting, and each dot in the figures represents mean accuracy of 3 network instances. Note that the blue line denotes the indicator of the efficiency of weight parameters or FLOPs in terms of accuracy. The upper left part from the blue line is the superior region while lower right part from blue line is the inferior region compared to the baseline models.

In Figure 3, we applied our optimal block (i.e. (d) block in Figure 2) on high- , middle- and low-level blocks, respectively. In our experiments, we evaluate the performance of the networks depending on the number of blocks where the proposed optimal block is applied. The model that we have tested is denoted as (transform type)-(# of the proposed blocks)-(hierarchy level in Low (L), Middle (M), and High (H) where the proposed optimal block is applied). For example, DWHT-3-L indicates the neural network model where the first three blocks in ShuffleNet-V2 consist of the proposed blocks, while the other blocks are the original blocks of ShuffleNet-V2. It is noted that in this experiment, we fix all the blocks with stride = 2 in the baseline model to be original ShuffleNet-V2 (Ma et al., 2018) stride = 2 blocks.

Figure 3 shows the performance of the proposed methods depending on the transform types {DCT, DWHT}, hierarchy levels {L, M, H} and the number of the proposed blocks that replace the original

ones in the baseline {3, 6, 10} in terms of Top-1 accuracy and the number of learnable weight parameters (or FLOPs). It is noted that, since the baseline model has only 7 blocks in the middle-level Stage (i.e. Stage3), we performed the middle-level experiments only for DCT/DWHT-3-M and -7-M models where the proposed blocks are applied from the end of Stage3 in the baseline model. In Figure 3, the performance of our 10-H (or 10-L), 6-H (or 6-L), 3-H (or 3-L) models (7-M and 3-M only for middle-level experiments) is listed in ascending order of the number of learnable weight parameters and FLOPs.

As shown in the first column of Figure 3, applying our optimal block on the high-level blocks achieved much better trade-off between the number of learnable weight parameters (FLOPs) and accuracy. Meanwhile, applying on middle- and low-level features suffered, respectively, slightly and severely from the inefficiency of the number of weight parameters (FLOPs) with regard to accuracy. This tendency is shown similarly for both DWHT-based models and DCT-based models, which implies that there can be an optimal hierarchical level of blocks favored by conventional transforms. Also note that our DWHT-based models showed slightly higher or same accuracy with less FLOPs in all the hierarchy level cases compared to our DCT-based models. This is because the fast version of DWHT does not require any multiplication but needs less amount of addition or subtraction operations compared to the fast version of DCT while it also has the sufficient ability to extract cross-channel information with the exquisite wiring-based structure.

For verifying the generality of the proposed method, we also applied our methods into MobileNet-V1 (Howard et al., 2017). Inspired by the above results showing that optimal hierarchy blocks for conventional transforms can be found in the high-level blocks, we replaced high-level blocks of baseline model (MobileNet-V1) and changed the number of proposed blocks that are replaced to verify the effectiveness of the proposed method. The experimental results are described in Table 2. Remarkably, as shown in Table 2, our DWHT-6-H model yielded the 1.49% increase in Top-1 accuracy even under the condition that the 79.4% of parameters and 49.4% of FLOPs are reduced compared with the baseline 1x model. This outstanding performance improvement comes from the depthwise separable convolutions used in MobileNet-V1, where PC layers play dominant roles in computation costs and memory space, i.e. they consume 94.86% in FLOPs and 74% in the total number of parameters in the whole network (Howard et al., 2017). The full performance results for all the hierarchy levels {L, M, H} and the number of blocks {3, 6, 10} (exceptionally, {3, 7} blocks for the middle level experiments) are described in Appendix A.

In Appendix A, based on the comprehensive experiments it can be concluded that i) the proposed PC block always shows its better efficiency of number of parameters and FLOPs when applied on high-levels compared to when applied on low-level in the network hierarchy; ii) the performance gain start to decrease when the number of transform based PC blocks exceeded a certain capacity of the networks.

| Model | Top-1 Acc (%) | # of Weights (ratio) | # of FLOPs (ratio) |
|---|---|---|---|
| Baseline | $67.15 \pm 0.3$ | 3.31M (1x) | 92.4M (1x) |
| DWHT-3-H | $68.19 \pm 0.35$ | 1.47M (0.44x) | 71.6M (0.77x) |
| DCT-3-H | $68.21 \pm 0.19$ | 1.47M (0.44x) | 72M (0.78x) |
| DWHT-6-H | $68.65 \pm 0.27$ | 0.68M (0.2x) | 46.7M (0.5x) |
| DCT-6-H | $67.95 \pm 0.53$ | 0.68M (0.2x) | 47.7M (0.51x) |

Table 2: Performance result of hierarchically applying our optimal block on CIFAR100 dataset. All the models are based on MobileNet-V1 with width hyper-parameter 1x. We replaced both stride 1, 2 blocks in the baseline model with the optimal block that consist of [$3 \times 3$ depthwise convolution - Batch Normalization - ReLU - CTPC - Batch Normalization] in series.

## 4 EXPERIMENTS AND ANALYSIS

In this section, we analyze the significant accuracy degradation of applying ReLU after our proposed PC layer. Additionally, we analyze the active utilization of 3x3 depthwise convolution weight kernel values which takes an auxiliary role for conventional transform being non-learnable.

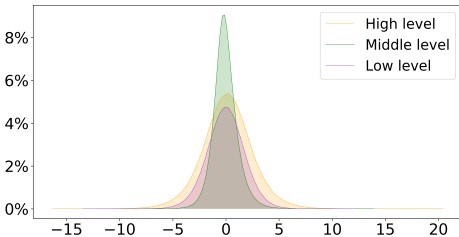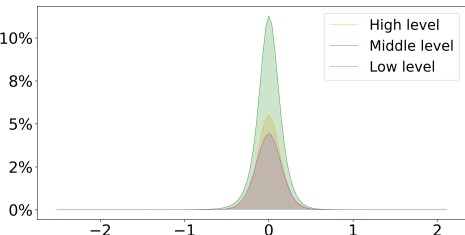

Figure 4: Histograms of hierarchy level (low-level, middle-level, high-level) activations after the proposed PC layer based on conventional transforms, Left: DWHT, Right: DCT. Both DWHT and DCT models are based on ShuffleNet V2 1.1x model where we replaced all of stride 1 blocks with (d)-DWHT w/o ReLU and (d)-DCT w/o ReLU blocks, respectively in Figure 2.

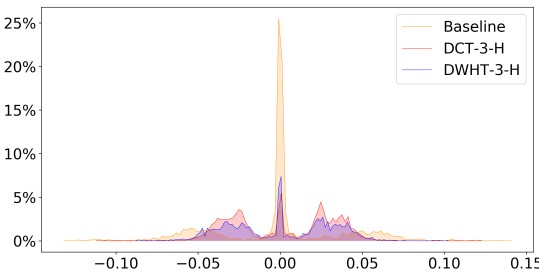

Figure 5: Histogram of $3 \times 3$ depthwise convolution weights in the third block, out of last 3 blocks. DCT-3-H and DWHT-3-H models are based on ShuffleNet V2 1.1x model with (d) block. Baseline model is ShuffleNet V2 1.1x model.

### 4.1 HINDRANCE OF RELU IN CROSS-CHANNEL REPRESENTABILITY

As shown in Table 1, applying ReLU after conventional transforms significantly harmed the accuracy. This is due to the properties of conventional transform basis kernels that both $H_m^D$ in Eq. 2 and $C_m$ in Eq. 1 have the same number of positive and negative parameters in the kernels except for $m = 0$ and that the distributions of absolute values of positive and negative elements in kernels are almost identical. These properties let us know that the output channel elements that have under zero value should also be considered during the forward pass; when forwarding $X_{ij}$ in Eq. 3 through the conventional transforms if some important channel elements in $X_{ij}$ that have larger values than others are combined with negative values of $C_m$ or $H_m^D$, the important cross-channel information in the output $Z_{ijm}$ in Eq. 3 can reside in the value range under zero. Figure 4 shows that all the hierarchy level activations from both DCT and DWHT based PC layer have not only positive values but also negative values in almost same proportion. These negative values possibly include important cross-channel correlation information. Thus, applying ReLU on activations of PC layers which are based on conventional transforms discards crucial cross-channel information contained in negative values that must be forwarded through, leading to significant accuracy drop as shown in the results of Table 1. Figure 6 empirically demonstrates above theoretical analysis by showing that as the negative value regions are fully ignored (i.e. $F =$ ReLU), the accuracy is significantly degraded while fully reflecting the negative value regions (i.e. $g = 1$) shows the best accuracy. From above kernel value based analysis and its experiment, we do not use non-linear activation function after the proposed PC layer.

### 4.2 ACTIVE $3 \times 3$ DEPTHWISE CONVOLUTION WEIGHTS

In Figure 5 and Appendix B, it is observed that $3 \times 3$ depthwise convolution weights of last 3 blocks in DWHT-3-H and DCT-3-H have much less near zero values than that of baseline model. That

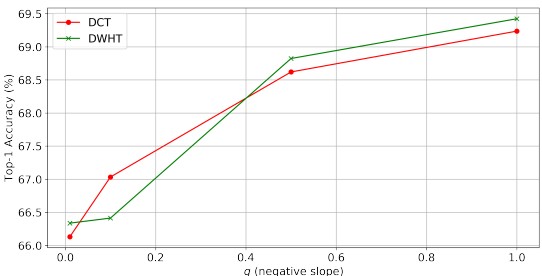

Figure 6: Ablation study of negative slope term $g$ in activation function $F$, which is defined as $F(x) = max(0, x) + g * min(0, x)$. The performance of models were evaluated based on DCT or DWHT-13-H ShuffleNet-V2 1.1x where we applied $F$ as an activation function after every DCT or DWHT based PC layer and Batch Normalization layer.

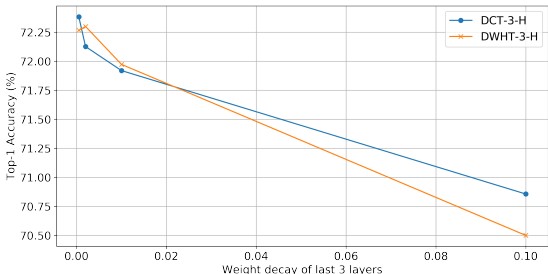

Figure 7: Ablation study of weight decay values (5e-4, 2e-3, 1e-2, 1e-1). We applied these weight decay values only on $3 \times 3$ depthwise convolution weights of last 3 blocks in DCT-based model and DWHT-based model, while all the other learnable weights were regularized with weight decay of 5e-4.

is, the number of values which are apart from near-zero is much larger on DCT-3-H and DWHT-3-H models than on baseline model. We conjecture that these learnable weights whose values are apart from near-zero were actively fitted to the optimal domain that is favored by conventional transforms. Consequently, these weights are actively and sufficiently utilized to take the auxiliary role for conventional transforms which are non-learnable, deriving accuracy increase compared to the conventional PC layer as shown in Figure 3.

To verify the impact of activeness of these $3 \times 3$ depthwise convolution weights in the last 3 blocks, we experimented with regularizing these weights varying the weight decay values. Higher weight decay values strongly regularize the scale of $3 \times 3$ depthwise convolution weight values in the last 3 blocks. Thus, strong constraint on the scale of these weight values hinders active utilization of these weights, which results in accuracy drop as shown in Figure 7.

## 5 CONCLUSION

We propose the new PC layers through conventional transforms. Our new PC layers allow the neural networks to be efficient in complexity of computation and learnable weight parameters. Especially for DWHT-based PC layer, its floating point multiplication-free property enabled extremely efficient in computation overhead. With the purpose of successfully fusing our PC layers into neural networks, we empirically found the optimal block unit structure and hierarchy level blocks in neural networks for conventional transforms, showing accuracy increase and great representability in cross-channel correlations. We further intrinsically revealed the hindrance of ReLU toward capturing the cross-channel representability and the activeness of depthwise convolution weights on the last blocks in our proposed neural network.

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

# A   GENERALITY OF APPLYING PROPOSED PC LAYERS IN OTHER NEURAL NETWORKS

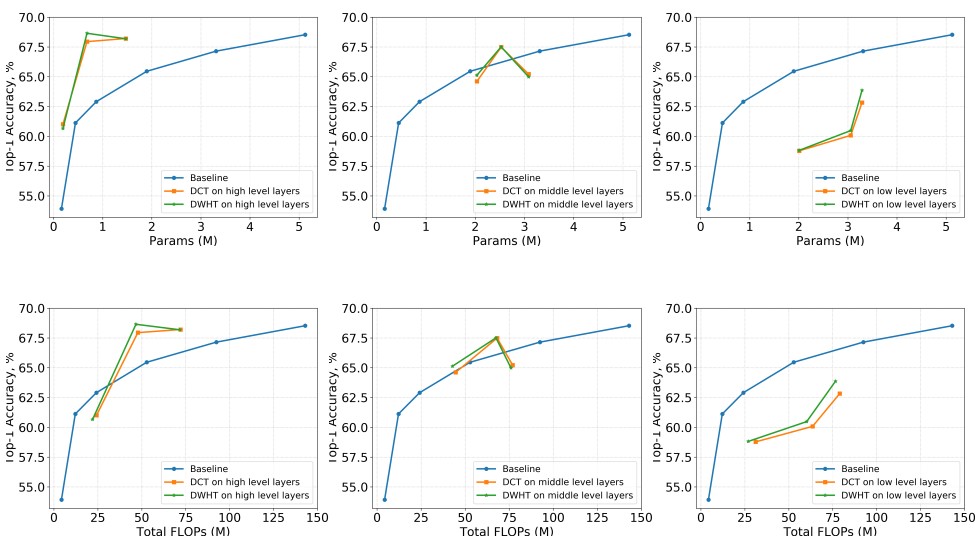

Figure 8: Performance curve of hierarchically applying our optimal block (See Table 2 for detail settings) on CIFAR100, Top: in the viewpoint of the number of learnable weight parameters, Bottom: in the viewpoint of the number of FLOPs. The performance of baseline models was evaluated by MobileNet-V1 architecture with width hyper-parameter 0.2x, 0.35x, 0.5x, 0.75x, 1x, 1.25x. Our proposed models were all experimented with 1x setting, and each dot in the figures represents mean accuracy of 3 network instances. Our models are experimented with 10-H, 6-H, 3-H models (first column) , 7-M, 3-M-Rear, 3-M-Front models (second column) and 10-L, 6-L, 3-L models (final column), listed in ascending order of the number of learnable weight parameters and FLOPs.

In Figure 8, for the purpose of finding more definite hierarchy level of blocks favored by our proposed PC layers, we subdivided our middle level experiment scheme; DCT/DWHT-3-M-Front model denotes the model which applied the proposed blocks from the beginning of Stage3 in the baseline while DCT/DWHT-3-M-Rear model denotes the model which applied from the end of Stage3. The performance curves of all our proposed models in Figure 8 show that if we apply the proposed optimal block within the first 6 blocks in the network, the Top-1 accuracy is mildly or significantly deteriorated compared to the required computational cost and number of learnable parameters, informing us the important fact that there are the definite hierarchy level blocks which are favored or not favored by our proposed PC layers in the network.

## B   HISTOGRAM OF $3 \times 3$ DEPTHWISE CONVOLUTION WEIGHTS IN HIGH-LEVEL BLOCKS

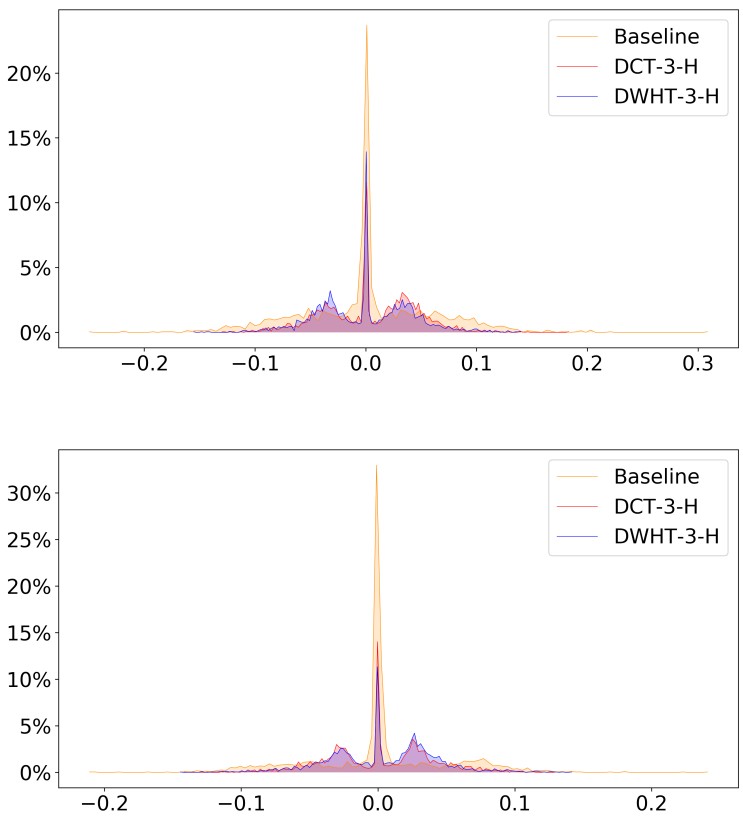

Figure 9: Histograms of $3 \times 3$ depthwise convolution weights, Top: histogram of first block out of last 3 blocks, Bottom: histogram of second block out of last 3 blocks. DWHT-3-H and DCT-3-H models are based on ShuffleNet-V2 1.1x model with (d)-DWHT w/o ReLU and (d)-DCT w/o ReLU block in Figure 2, respectively. Baseline model is ShuffleNet-V2 1.1x model.

## C   PERFORMANCE COMPARISON BETWEEN RCPC AND PROPOSED PC LAYERS

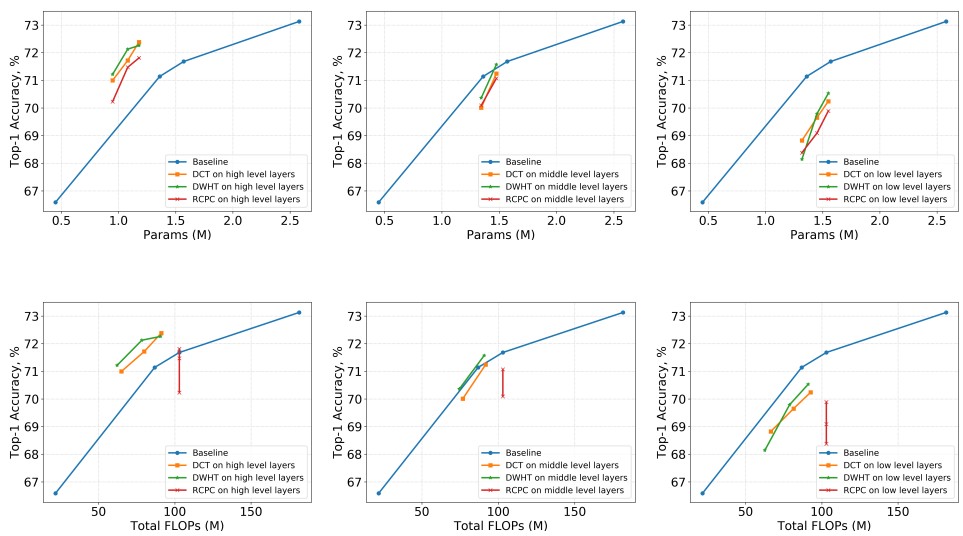

Figure 10: Performance curve of hierarchically applying our proposed DCT/DWHT based PC layers and RCPC layer on CIFAR100 dataset. Experimental settings are the same as described in Figure 3. Our DCT/DWHT-based models are experimented with 10-H, 6-H, 3-H models (first column) , 7-M, 3-M models (second column) and 10-L, 6-L, 3-L models (final column), listed in ascending order of the number of learnable weight parameters and FLOPs. Models equipped with RCPC layers are experimented with 10-H, 6-H, 3-H models (first column), 7-M, 3-M models (second column) and 10-L, 6-L, 3-L models (final column), listed in ascending order of number of learnable parameters (top row) and Top-1 accuracy (bottom row).

For the purpose of demonstrating the superiority of our proposed DCT/DWHT based PC layers over RCPC layer in all the hierarchical (i.e. low/mid/high) level layers, we compared the performance trade-off in Figure 10. It is noted that DCT/DWHT based PC layers almost always get higher accuracy than RCPC layer in all the hierarchical level layers. Comparing the distance between the orange or green line with the red line in Figure 10, our DCT/DWHT based PC layers showed high efficiency in trade-off between accuracy and the computational costs or number of learnable parameters, compared to RCPC layer in almost all the hierarchical levels.

# D  GENERALITY OF APPLYING OUR PROPOSED PC LAYERS ON OTHER TASKS

| Model | Easy (mAP %) | Medium (mAP %) | Hard (mAP %) | # of Weights (ratio) | # of FLOPs (ratio) |
|-------|--------------|----------------|--------------|----------------------|--------------------|
| Baseline | 89.74 | 86.3 | 64.11 | 2.32M (1x) | 129.4M (1x) |
| DWHT-3-H | 89.82 | 86.27 | 64.38 | 0.48M (0.2x) | 108.6M (0.84x) |
| DCT-3-H | 90.25 | 86.83 | 61.49 | 0.48M (0.2x) | 109M (0.84x) |
| DWHT-6-H | 89.98 | 86.29 | 63.63 | 0.25M (0.11x) | 95.1M (0.73x) |
| DCT-6-H | 90.11 | 86.71 | 62.16 | 0.25M (0.11x) | 96.3M (0.74x) |

Table 3: Quantitative comparison between the baseline model and our DCT/DWHT-based models on WIDER FACE validation dataset.

| Model | AP (%) | # of Weights (ratio) | # of FLOPs (ratio) |
|-------|--------|----------------------|--------------------|
| Baseline | 94.41 | 2.32M (1x) | 129.4M (1x) |
| DWHT-3-H | 94.5 | 0.48M (0.2x) | 108.6M (0.84x) |
| DCT-3-H | 93.79 | 0.48M (0.2x) | 109M (0.84x) |
| DWHT-6-H | 94.37 | 0.25M (0.11x) | 95.1M (0.73x) |
| DCT-6-H | 93.67 | 0.25M (0.11x) | 96.3M (0.74x) |

Table 4: Quantitative comparison between the baseline model and our proposed DCT/DWHT-based models on FDDB dataset. AP means the true positive rate at 1,000 false positives and all the models were evaluated with discontinuous criterion in FDDB dataset.

In order to demonstrate the domain-generality of the proposed method, we conducted comprehensive experiments on applying our proposed PC layers to object detection, specifically to the face detection task. For face detection schemes such as anchor design, data augmentation and feature-map resolution design, we followed Zhang et al. (2017a) which is one of the baseline methods in face detection field. It is noted that there is a huge demand on real-time face detection algorithms having high detection accuracy, which leads us to applying our PC layers to a lightweight face detection network. Therefore, instead of using VGG16 (Simonyan & Zisserman, 2014) as backbone network as in Zhang et al. (2017a), we set MobileNet-V1 0.25x as our baseline backbone model where extra depthwise separable blocks are added for detecting more diverse scales of face in the images. In this baseline model, we replaced the conventional PC layers within last 3, 6 blocks with our DCT/DWHT based PC layers. We trained all the models on the WIDER FACE (Yang et al., 2016) train dataset and evaluated on WIDER FACE validation dataset and Face Detection Data Set and Benchmark (FDDB) dataset (Jain & Learned-Miller, 2010). WIDER FACE validation set has Easy, Medium and Hard subsets, which correspond to large, medium and small scale faces, respectively. Validation results of the baseline model and our proposed DCT/DWHT models on WIDER FACE are described in Table 3.

In Table 3, we note that, overall, our DWHT-3-H and DWHT-6-H models showed comparable or even higher mAP values than the baseline model on all the subsets (Easy, Medium, and Hard) with significantly reduced number of learnable parameters and FLOPs. Especially, DWHT-3-H model achieved 0.27% higher mAP than the baseline model under the condition that 79% of parameters and 16% of FLOPs are reduced on Hard subset. Regarding DCT-3-H and DCT-6-H models, they showed a solid improvement of mAP on Easy and Medium subsets with significantly reduced number of parameters and FLOPs compared to the baseline model.

Additionally, we verified the effectiveness of the proposed method on the FDDB dataset in Table 4. We note that our DWHT-6-H and DWHT-3-H models showed comparable or even 0.09% higher AP than the baseline model with significantly reduced number of learnable parameters and FLOPs. On the other hand, our DCT-6-H and DCT-3-H models showed a small degree of degradation on AP compared to the baseline model, which is a mild degradation considering the reduced amount of parameters and FLOPs.

Consequently, our comprehensive experiments on both WIDER FACE and FDDB datasets reveal the generality of our proposed method, enabling neural networks to be extremely lightweight and reduce the computational overhead.

