# OpenReview forum: "A NEW POINTWISE CONVOLUTION IN DEEP NEURAL NETWORKS THROUGH EXTREMELY FAST AND NON PARAMETRIC TRANSFORMS"
_ICLR.cc/2020/Conference — Reject_

### Official Review · AnonReviewer2 · 2019-10-23
**Official Blind Review #2**

**Rating:** 3

**Review:**

This paper proposes a new pointwise convolution layer, which is non-parametric and can be efficient thanks to the fast conventional transforms. Specifically, it could use either DCT or DHWT to do the transforming job and explores the optimal block structure to use this new kind of PC layer. Extensive experimental studies are provided to verify the new PC layer and experimental results show that the new layer could reduce the parameters and FLOPs while not loosing accuracy.

Overall, this paper provides a promising PC option for the CV community and the experiments seems solid. Although the novelty is limited, which is just to combines the DCT/DHWT with NN, the experiments are sufficient and I am glad to see that this simple idea works in practice.

Concerns: This paper is based on the practical validation. Is there any theory to support that using DCT/DHWT could achieve better performance. In Image Processing, DCT/DHWT could be used to compress images or videos, but what is the benefit to use them in computer vision?

---post comments after rebuttal---
Thanks for the response.

However, the rebuttal does not address my concerns. The logic behind the authors response is that since there is much work that uses DCT / DHWT-like features in computer vision tasks, we can also use them to replace the conventional PC layers. Unfortunately, I do not think that is the theory to support the idea described in this paper. Actually, the success of CNN architecture has already confirmed the advantage of conventional PC layer, it seems it is a backward step for the community. Moreover, I agree with Reviewer#4 that ImageNet should be used as benchmark to so the advantage of the paper.

Hence, after reading the response and the reviews from other reviewers, I support to reject the work.

**Experience Assessment:**

I have read many papers in this area.

**Review Assessment: Checking Correctness Of Derivations And Theory:**

I assessed the sensibility of the derivations and theory.

**Review Assessment: Checking Correctness Of Experiments:**

I assessed the sensibility of the experiments.

**Review Assessment: Thoroughness In Paper Reading:**

I read the paper at least twice and used my best judgement in assessing the paper.

---

> ### Author Response · Authors · 2019-11-12
> **Response to Reviewer #2**
>
> We sincerely appreciate the reviewer for valuable and intuitive comments.
>
> Regarding the benefit of using the DCT / DHWT in computer vision tasks, they have often been used to extract perceptually meaningful features from raw (RGB) image input. For instance, Viola and Jones [1] proposed a very fast face detection algorithm where Haar-like features (a variant of DWHT for the spatial shapes) are crucially used to characterize distinctive facial regions in contrast (e.g., eye and glabella regions). Also, DCT / DWHT basis functions have been used as template functions in template matching [2, 3], since the DCT / DWHT basis functions contain various spatial patterns with different spatial frequency, thus capturing both fine details (high frequency) as well as base shapes (low frequency) in an input image.
> Furthermore, DCT and DWHT have been used for dimensionality reduction in diverse domains such as face recognition [4], speech coding [5] and matrix approximation [6], due to their sparse representation property and information preservation property (i.e., DCT/DWHT can well approximate the original signals using only a limited number of coefficients).
> Relying on the aforementioned nice properties of DCT/DHWT, we propose to replace the conventional PC layers with traditional DCT / DWHT transforms in neural networks. Our comprehensive experimental results in Figure 3 and Table 2 show that the proposed PC layers achieve better trade-off between accuracy and computational complexity than the conventional PC layer. These experimental results clearly indicate that lack of flexibility by learning the weights in the conventional PC layer can be effectively overcome by pre-defined weights if the weight kernels have the good properties of extracting features and preserving information.
>
> [1] Viola, Paul, and Michael Jones. "Rapid object detection using a boosted cascade of simple features." In CVPR (1) 1.511-518 (2001): 3.
> [2] Hörhan, Markus, and Horst Eidenberger. "An efficient DCT template-based object detection method using phase correlation." In 2016 50th Asilomar Conference on Signals, Systems and Computers. IEEE, 2016.
> [3] Wang, Song, and Jiankun Hu. "A Hadamard transform-based method for the design of cancellable fingerprint templates." In 2013 6th International Congress on Image and Signal Processing (CISP). Vol. 3. IEEE, 2013.
> [4] Amine, Aouatif, et al. "Investigation of feature dimension reduction based DCT/SVM for face recognition." In 2008 IEEE Symposium on Computers and Communications. IEEE, 2008
> [5] Xu, Jingyun, et al. "DCT based algorithm on dimension reduction of residual frequency magnitude parameters." In 2014 International Conference on Audio, Language and Image Processing. IEEE, 2014.
> [6] Tropp, Joel A. "Improved analysis of the subsampled randomized Hadamard transform." In Advances in Adaptive Data Analysis 3.01n02 (2011): 115-126.

---

### Official Review · AnonReviewer1 · 2019-10-23
**Official Blind Review #1**

**Rating:** 8

**Review:**

Summary:

This paper proposes using non-parametric filters like Discrete Cosine Transform (DCT) and Discrete Walsh-Hadamard Transform (DWHT) which have been widely used as feature extractors in vision and image processing before deep learning became prevalent as layers especially to replace pointwise convolution (PC) layers in deep network architectures like ShuffleNet-v2 and MobileNet-V1.

The motivation is that using such layers can capture cross-channel correlations without addition of extra parameters that need to be learnt and by replacing PC layers which tend to make up the bulk of the parameters in such settings large reduction in number of weights and flops can be achieved with little drop (or even increase) in accuracy.

They show experiments on cifar100 datasets.

Comments:

- The paper is overall easy to read although the writing and presentation can use some work.

- I really enjoyed reading the paper though because it seems in the deep learning era we have a tendency to re-learn what we already know. This paper shows that combining our knowledge of image processing and compressed sensing with good function approximation leads to better and more compact representation learning. I think these aspects are being generally overlooked currently but wont be surprised to see more of these papers.

- While reading section 3.2 it occurred to me that it might be interesting to consider throwing these layers in to a neural architecture search (NAS) algorithm and let it figure out the right architecture. (Just a suggestion for future work not asking for this in the rebuttal.)

- Overall I am positive about the paper and have no major asks but if time and resources permit perhaps trying out the same experiments on ImageNet to see if the trend holds.

**Experience Assessment:**

I have published in this field for several years.

**Review Assessment: Checking Correctness Of Derivations And Theory:**

I assessed the sensibility of the derivations and theory.

**Review Assessment: Checking Correctness Of Experiments:**

I assessed the sensibility of the experiments.

**Review Assessment: Thoroughness In Paper Reading:**

I read the paper at least twice and used my best judgement in assessing the paper.

---

> ### Author Response · Authors · 2019-11-11
> **Response to Reviewer #1**
>
> As a future work, we will indeed research for applying Neural Architecture Search (NAS) in our proposed PC layers. From a microscopic viewpoint, we expect NAS would help to find new DCT, DWHT blocks rather than just applying our DCT / DWHT on human-designed blocks (ShuffleNet blocks, MobileNet blocks). These optimal block structures discovered by NAS might enable our DCT / DWHT based PC layers to more sufficiently represent the correlation between the feature maps while taking more advantages of light-weight and low-computational overhead. Meanwhile from a macroscopic viewpoint, we expect NAS can optimize the network to find the position where the proposed PC layers can be applied under the condition of improving the classification accuracy, which can reduce the time and burden of human researchers to apply our PC layers. Thus this future work will be valuable for introducing generality of our method. Additionally, regarding the experiments on ImageNet, we leave them as future works due to the limited time and resources. We sincerely thank the reviewer for the insightful suggestion and comments.

---

### Official Review · AnonReviewer4 · 2019-11-01
**Official Blind Review #4**

**Rating:** 3

**Review:**

This paper presents a new pointwise convolution (PC) method which applies conventional transforms such as DWHT and DCT. The proposed method aims to reduce the computational complexity of CNNs without degrading the performance. Compared with the original PC layer, the DWHT/DCT-based methods do not require any learnable parameters and reduce the floating-point operations. The paper also empirically optimizes the networks by removing ReLU after the proposed PC layers and using conventional transforms for high-level features extraction. Experiments on CIFAR100 show that the DWHT-based model improves the accuracy and reduces parameters and FLOPs compared with MobileNet-V1.

Although this paper is well organized and easy to follow, the novelty of the proposal seems limited and the performance improvement claimed by the author(s) is not very convincing due to the insufficiency of experiments. Firstly, the proposed method is just a manually designed and fixed 1*1 convolutional kernel, and its superiority over random initialization seems very limited as shown in Table 1. Also, the proposed method makes accuracy degrade when applied to low- and middle-level features. I wonder whether there is a more theoretical explanation for that. Moreover, the experiments are performed only on a small dataset CIFAR100. According to my own experience, the artificial convolutional kernels with some prior knowledge may work well on small datasets but tend to fail on larger ones. More experiments on larger-scale datasets like ImageNet are recommended to make results more convincing.

Therefore, my decision leans to a rejection.

Some questions:
1. How is the performance when applying RCPC only to low-/middle-/high-level features? I suppose it should be proved that the proposed method is definitely better than random initialization.
2. Why is only applying the proposed block to high-level layers working? How is the trade-off between parameters and accuracy different for each level of features?

Spelling mistake:
Page 6: in the second last paragraph, 'non-parameteric' should be 'non-parametric'.

**Experience Assessment:**

I have published one or two papers in this area.

**Review Assessment: Checking Correctness Of Derivations And Theory:**

I assessed the sensibility of the derivations and theory.

**Review Assessment: Checking Correctness Of Experiments:**

I carefully checked the experiments.

**Review Assessment: Thoroughness In Paper Reading:**

I read the paper thoroughly.

---

> ### Author Response · Authors · 2019-11-14
> **Response to Reviewer #4 (part 1)**
>
> 1. How is the performance when applying RCPC only to low-/middle-/high-level features?
> Following the reviewer’s comment, we performed comprehensive experiments to see the performance of RCPC depending on its locations of low-/middle-/high-level levels. Table R-1 shows that the proposed PC layers (DCT and DWHT) outperform RCPC in terms of accuracy in almost all of the low-/middle-/high-levels under lower computation complexity (i.e., FLOPs) condition. This clearly indicates that the proposed PC layers can have better feature representation performance than RCPC. (For better visibility, we plotted trade-off graph in Appendix C of the revised manuscript.)
>
> Table R-1. Performance comparison among baseline, the proposed PC layers (DCT and DWHT), and RCPC depending on different layer positions.
>
>  |Model|Top-1 Acc (%)|# of weights|# of FLOPs|
>
>  |Baseline (ShuffleNet-V2 1.1x) |71.68|1.57M|102.9M|
>
>  | RCPC-3-H |71.81|1.17M|102.9M|
>  | DCT-3-H   |72.38|1.17M|91.1M |
>  | DWHT-3-H|72.26|1.17M|90.5M|
>
>  | RCPC-6-H |71.47|1.08M|102.9M|
>  | DCT-6-H   |71.71|1.08M|79.9M|
>  | DWHT-6-H|72.12|1.08M|78.3M|
>
>  | RCPC-10-H |70.23|0.94M|102.9M|
>  | DCT-10-H    |70.99|0.94M |64.9M|
>  | DWHT-10-H |71.21|0.94M|62.1M|
>
>  | RCPC-3-M |71.07|1.47M|102.9M|
>  | DCT-3-M   |71.24|1.47M|91.7M|
>  | DWHT-3-M |71.57|1.47M|90.7M|
>
>  | RCPC-7-M |70.09|1.34M|102.9M|
>  | DCT-7-M    |70.01|1.34M|76.7M|
>  | DWHT-7-M |70.36|1.34M|74.4M|
>
>  | RCPC-3-L |69.89|1.54M|102.9M|
>  | DCT-3-L    |70.24|1.54M|92.6M|
>  | DWHT-3-L |70.53|1.54M|91M|
>
>  | RCPC-6-L |69.09|1.45M|102.9M|
>  | DCT-6-L    |69.65|1.45M|81.4M|
>  | DWHT-6-L |69.79|1.45M|78.8M|
>
>  | RCPC-10-L |68.38|1.31M|102.9M|
>  | DCT-10-L    |68.82|1.31M|66.5M|
>  | DWHT-10-L |68.14|1.31M|62.6M|
>
> Additionally, in Table 1 of the original manuscript, there are some mistakes in putting values, which are corrected in the revised manuscript. (In case of modified accuracy values, we release source code for reproducing the accuracy results. Please see the reponse to all the reviewers for more detail.)  Table 1 in the revised manuscript clearly shows that, although the proposed PC layers take approximately only half of FLOPS compared to RCPC, the DWHT and DCT based PC layers achieve 1.26% and 1.07% higher top-1 accuracy than RCPC, respectively, which is a solid improvement in CIFAR100.
>
> 2. Why is only applying the proposed block to high-level layers working?
> In the viewpoint of the information theory in [1], Figure 7 in [1] demonstrates that, in general, the quantity of information for the input data  $X$ (i.e.,  $I(X; T)$, where $T$ denotes the hidden states) of the low level layers is high regardless of the number of training samples. This means that the learned kernels of low level layers maximally preserve the input information. Therefore, if we enforce these learnable kernels to be fixed with random and/or the DCT/ DWHT kernels during training, $I(X; T)$ of low level layers will be significantly lower, which serves as a information bottleneck that prevents rich information flow to high level layers.
>
> Meanwhile, on the high level layers, [1] observed that high level layers keep more information on the desired output variable $Y$ while compressing the irrelevant information in $X$. Likewise, since DCT and DWHT have a functionality to compress input signals by representing sparse coefficients, they enable our proposed PC layers to successfully replace the roles of the original high-level PC layers to compress irrelevant information in X, resulting in higher compression ratio of deep neural networks.
>
> 3. How is the trade-off between parameters and accuracy different for each level of features?
> The baseline models we employed are MobileNet-V1, ShuffleNet-V2 in which the number of channels increases as the depth of layers increases, ranging from 32 to 1024 (e.g. MobileNet-V1). To be specific, the conventional PC layers at  low levels  have small number of input/output channels (e.g. 32 / 64) while high level layers have large number of input/output channels (e.g. 512 / 1024). In such settings, it is definitely clear that high level PC layers have much more parameters (e.g. 512 * 1024) than low level PC layers (e.g. 32 * 64). Thus replacing conventional PC layers with our non-parametric PC layers in high level layers enjoyed the accuracy gain while the number of parameters are largely reduced, as shown in Figure 3 and Figure 8 of the original manuscript.
>
> Regarding the spelling mistake, we all corrected them in the revised manuscript.
>
> We sincerely thank the reviewer for closely reviewing our proposed PC layers and we will fully accommodate your insightful comments in the revised manuscript.
>
> [1] Shwartz-Ziv, Ravid, and Naftali Tishby. "Opening the black box of deep neural networks via information." In arXiv preprint arXiv:1703.00810 (2017).

---

> > ### Author Response · Authors · 2019-11-14
> > **Response to Reviewer #4 (part 2)**
> >
> > Additionally, due to time and resource limitation by the rebuttal deadline, it was not possible to conduct heavy experiments with 1M-scale hugh datasets such as ImageNet. However, we will perform the experiments on ImageNet as future work.
> > Instead, we conducted comprehensive experiments on applying our proposed PC layers to object detection, specifically to the face detection task in order to demonstrate the domain-generality of the proposed method. Detailed experimental settings and results are described in Appendix D of the revised manuscript.

---

### Author Response · Authors · 2019-11-14
**Response to all reviewers**

We sincerely appreciate the constructive and insightful comments from all the reviewers. We have revised the manuscript following the suggestions from the reviewers and the major changes are summarized as follows;

> We corrected the values that are mistakenly put and all the misspelled words.

> We re-plotted Figure 3 for the visibility of baseline 1.5x model.

> We experimented on applying RCPC on all the hierarchical level layers and added the trade-off figures in Appendix C of the revised manuscript.

> We experimented on applying DCT / DWHT based PC layers to the Face Detection task and added the results in Appendix D of the revised manuscript.

For clarity, we denoted all the major and informative changes including above with blue color in the revised manuscript. (In Appendix C and D, only the section name is denoted with blue for readability.)

Also, regarding the modified accuracy of the models (i.e. (c)-DWHT w/ ReLU and (d)-DWHT w/o ReLU models)  in Table 1 of the revised manuscript, we release the source code which reproduces the exact accuracy of those models in https://github.com/iclrpaper653/iclrpaper653_reprod

Please see responses to individual reviewer comments below, for more detailed information.

---

### Decision · Program_Chairs · 2019-12-19

**Decision:**

Reject

**Comment:**

This paper presents an approach to utilize conventional frequency domain basis such as DWHT and DCT to replace the standard point-wise convolution, which can significantly reduce the computational complexity. The paper is generally well-written and easy to follow. However, the technical novelty seems limited as it is basically a simple combination of CNNs and traditional filters. Moreover, as reviewers suggested, it is our history and current consensus in the community that learned representations have significantly outperformed traditional pre-defined features or filters as the training data expands. I do understand the scientific value of revisiting and challenging that belief as commented by R1, but in order to provoke meaningful discussion, experiments on large-scale dataset like ImageNet are definitely necessary. For these reasons, I think the paper is not ready for publication at ICLR and would like to recommend rejection.